THE NATURAL HISTORY OF MODEL ORGANISMS

# The Norway rat, from an obnoxious pest to a laboratory pet

**Abstract** The laboratory rat was the first mammal domesticated for research purposes. It is descended from wild Norway rats, *Rattus norvegicus*, which despite their name likely originated in Asia. Exceptionally adaptable, these rodents now inhabit almost all environments on Earth, especially near human settlements where they are often seen as pests. The laboratory rat thrives in captivity, and its domestication has produced many inbred and outbred lines that are used for different purposes, including medical trials and behavioral studies. Differences between wild Norway rats and their laboratory counterparts were first noted in the early 20[th] century and led some researchers to later question its value as a model organism. While these views are probably unjustified, the advanced domestication of the laboratory rat does suggest that resuming studies of wild rats could benefit the wider research community.

**KLAUDIA MODLINSKA\* AND WOJCIECH PISULA**

**\*For correspondence:**
kmodlinska@psych.pan.pl

**Competing interests:** The authors declare that no competing interests exist.

## Introduction

The Norway rat, *Rattus norvegicus*, is known by many names such as the brown rat, common rat, sewer rat, Hanover rat, Norwegian rat, city rat, water rat and wharf rat. Living in close proximity to humans, wild Norway rats are often considered pests (*Khlyap et al., 2012*). They are well known for invading and damaging property, spoiling food supplies and spreading diseases (*Kosoy et al., 2015*). Their seemingly unrestricted capacity to reproduce, their ferocious appetite (which can result in cannibalism) and their remarkable ability to survive in adverse and often unsanitary conditions only seem to worsen their reputation among many in the general public. For all of these reasons and more, rats are the targets of intensive pest control strategies.

In spite of their bad reputation in the wild, the laboratory rat is perhaps the archetypal model organism. Widely used in fields such as neuroscience, physiology and toxicology, 'lab rats' account for 13.9% of all animals used in research in Europe (*European commission, 2012*), second only to mice which account for 60.9%. First domesticated from wild Norway rats

over 170 years ago (*Richter, 1959*), today laboratory rats owe their popularity as a model organism largely due to their widespread availability, low breeding costs, short reproductive cycle and ability to thrive in captive environments.

Laboratory rats differ from Norway rats in the wild, just like many other model organisms (*Alfred and Baldwin, 2015*). In the mid-20[th] century, these differences led some researchers to suggest that the laboratory rat had become a degenerated form of its wild cousins and lost its value as a study model (*Beach, 1950*; *Lockard, 1968*). While these views are probably unjustified, researchers working with laboratory rats should remain aware of its advanced domestication. While few modern laboratories study wild *R. norvegicus* colonies, a better appreciation of the rat's natural history would expand its value as a model organism. Resuming studies of wild rats would give the opportunity to not only 'refresh' genetic lines and create new highly specialized strains, but also document the many changes that have taken place in wild

populations since most laboratory lines were first obtained.

## Natural history

*Rattus norvegicus* is one of over 60 species in the mammalian genus *Rattus* (*Musser and Carlton, 2005*), which can be divided into seven systematic groups (*Box 1*). The deepest divergence within the genus occurred 3.5 million years ago and separates a lineage of rats that are endemic to New Guinea from the other groups (*Robins et al., 2008*). Rats belong to the *Muridae* family in the *Rodentia* order. This family also includes mice (genus *Mus*), and rats and mice are thought to have diverged about 40 million years ago (*Adkins et al., 2001*).

Despite its name, the Norway rat most likely originated in Asia. It diverged from its sibling species the Himalayan field rat (*Rattus nitidus*) around 620–644 thousand years ago (*Teng et al., 2017*), and some of the oldest remains of *R. norvegicus* have been discovered in the Chinese province of Sichuan-Guizhou (*Musser and Carlton, 2005*). The Norway rat got its name as it was believed to have immigrated to England from Norway aboard ships in the 18th century. However, the species originally arrived in European countries from Asia via Russia, superseding the older black rat *Rattus rattus*. Numerous remains of the species have been discovered at archaeological sites dated to the 14th century (for instance, in Tarquinia, Italy), suggesting that small populations of these rats had actually inhabited Europe earlier than previously thought (*Clark et al., 1989*). The Norway rat reached North America between 1750 and 1775 (*Nowak, 1999*). Some places in northeast and central Asia were not inhabited by the Norway rat until the last decades of the 20th century (*Khlyap and Warshavsky, 2010*).

## Ecology

Rats live in almost all terrestrial environments except deserts, tundra and polar ice. They adapt easily to new conditions thanks to their physical resilience, omnivorous diet and flexible behavior. Like the black rat, the Norway rat often lives in the immediate vicinity of humans, including in cities (*Aplin et al., 2011*), and can pose a serious threat to human health because it may carry various pathogens and parasites (*Box 2*). Wild Norway rats often inhabit storage facilities, basements, deserted buildings and landfill sites where human-generated waste is deposited (*Sacchi et al., 2008*). In cities, its habitats are distributed irregularly, and each rat's home range is relatively restricted compared to rats in less urban settings. City rats prefer areas with rich vegetation, banks of water reservoirs, old buildings and sewer systems (*Ayyad et al., 2018*; *Traweger et al., 2006*; *van Adrichem et al., 2013*). They dig burrows and build

## Box 1. Systematics of the genus *Rattus*.

According to *Musser and Carlton (2005)*, the species belonging to the genus *Rattus* may be divided into seven groups:

- the "norvegicus" species group, including *R. norvegicus* and a few related species
- the "exulans" species group comprising only *Rattus exulans* (the Polynesian rat)
- the "rattus" group comprising *Rattus rattus* (black rat or roof rat), *Rattus tanezumi* (Tanezumi rat) and a large number of closely related species
- the native Australian group, including *Rattus fuscipes* (bush rat)
- the native New Guinean group including *Rattus leucopus* (Cape Cork rat) and *Rattus praetor* (large New Guinea spiny rat)
- the native Sulawesian group, including *Rattus xanthurus* (yellow-tailed rat)
- an uncertain "group" containing unaffiliated species whose phylogenetic history has not yet been established

extensive systems of tunnels and passages in riverbanks and open spaces, where they live and breed (*Barnett, 2005*). Like most mammals, rats are characterized by female philopatry and male dispersal (*Gardner-Santana et al., 2009*). Rats choose their habitats based on the availability of shelter, food and water (*Orgain and Schein, 1953*).

## Characteristics of the wild Norway rat

### Reproduction

Wild rats reach sexual maturity at about 11 weeks, remain pregnant for 21–24 days, and give birth to litters of about 7 or 8 pups. Female rats build nests before giving birth, and the young are born almost naked, blind and totally dependent on the mother (*Burton and Burton, 2002*). The young start leaving the nest and ingest solid foods at about 14 days after birth. *R. norvegicus* can breed all year long and has 3–5 litters per year on average. Its life expectancy is slightly more than 1 year (*Davis, 1953*).

### Behavior and senses

The Norway rat is primarily nocturnal. It prefers small, dark, confined places and avoids moving in open and well-lit spaces. It tends to move on four limbs with its fur and whiskers in contact with the walls and large objects. It can also jump (*Himmler et al., 2014*), and swim and dive (*Galef, 1980*; *Stryjek et al., 2012*). Rats have no sweat glands and regulate their body temperature through behavior, for example, by hiding in burrows. The sparsely haired tail also plays a part in thermoregulation (*Owens et al., 2002*).

In rats, the main sensory input is touch from the facial whiskers (or vibrissae) and a particularly well-developed sense of smell (*Uchida and Mainen, 2003*). Wild Norway rats have relatively poor eyesight and are sensitive to sharp light (*Finlay and Sengelaub, 1981*; *Prusky et al., 2002*). They have dichromatic color vision thanks to two classes of cone cells on the retina: one sensitive to ultraviolet light and the other most sensitive to the middle wavelengths of the visible spectrum, such as the color green (*Jacobs et al., 2001*). They can detect sounds between about 0.25–80 KHz (*Heffner et al., 1994*), which enables them to communicate with ultrasound (*Portfors, 2007*; *Burke et al., 2018*). These vocalizations are inaudible to humans without the use of specialized equipment.

### Exploration and neophobia

Rats are highly inquisitive and eager to explore new environments but exhibit neophobia (i.e., caution towards new objects; *Pisula, 2009*). They also markedly reduce their food intake after they are introduced to an unfamiliar food. This "food neophobia" is typified by the initial avoidance of the new food, followed by gradual sampling (*Barnett, 1958*). If the new food does not become associated with adverse body symptoms, the rats will eat more (*Barnett, 2009*; *Mitchell, 1976*). Rats develop an aversion to foods that cause adverse effects within up to 6 hours (*Misanin et al., 2002*; *Revusky and Bedarf, 1967*), which often limits the effectiveness of traditional pest control procedures.

### Social behavior

Rats live in groups and establish social relations. In favorable conditions they can form colonies of several hundred individuals. The colonies comprise groups with an adult male and a few females with their young. These groups inhabit certain areas, called territories, which are delineated and marked with scent cues (*Adams, 1976*; *Barnett, 2009*). The males defend their territories against intruders from other groups (referred to "resident-intruder aggression"; *Koolhaas et al., 2013*). Social aggression in males may increase while cohabiting with females (*Albert et al., 1988*). When individual rats meet, they examine each other thoroughly, relying on scent to learn about the sex, age, health, reproductive status and nutrition of the other rat. If an individual is not recognized as a representative of its own group, the intruder may be attacked and will often retreat from the territory (*Miczek and de Boer, 2005*). Female rats defend their nests and offspring against intruders and their social aggression increases in the postpartum period (*Consiglio and Lucion, 1996*).

Juvenile rats engage in play-fighting (*Pellis and Pellis, 1987*). Rats in the same group groom each other, sleep in tight groups and huddle. The group also provides a setting for rats to learn from each other about food sources and food quality. Rats develop preferences for particular foods by sniffing at the mouth and fur of an individual who has finished eating (*Galef, 1993*). There is no evidence that aversion to foods that have made a specific individual sick is transmitted from one individual to the next.

## Box 2. Disease and pest control.

Wild Norway rats are commonly perceived as dirty animals, inhabiting sewage systems and feeding on garbage. While the reality is that rats are fastidiously clean animals that groom themselves several times a day, they are nonetheless vectors of numerous diseases. Bacterial infections can spread from rats to humans via multiple routes, including rat bites or contact with the animal's urine (*Himsworth et al., 2013*). Other bacteria are transmitted from rats to humans by fleas (*Civen and Ngo, 2008*). These include bacteria in the genus *Yersinia* which cause bubonic plague. *Yersinia* bacteria are present in wild rat populations inhabiting cities in Africa, southeast Asia, and South America (*Boey et al., 2019*). However, contrary to popular belief, it was the black rat and not the Norway rat that was most likely responsible for the pandemic outbreak of bubonic plague that occurred in the 14th century. Rats are also an important source of antimicrobial resistant bacteria which may infect humans and other animals (*Gakuya et al., 2001*), and they are the primary reservoir of a hantavirus known as Seoul virus, which causes a hemorrhagic fever with renal syndrome in humans (*Jonsson et al., 2010*).

Due to the disease risk they represent (and the material damage they can cause), humans have strived to eliminate rats from their settlements for centuries. Today the most commonly used pest control methods include traps, rodenticides, biological control, reproductive inhibition and ultrasonic devices (*Tobin and Fall, 2004*). Older toxic compounds – such as sodium fluoroacetate, strychnine and zinc phosphide – are still used but have limited efficacy for large populations and long-term campaigns. Rats quickly develop strong aversion to the taste of substances which have caused illness (*Riley and Tuck, 1985*). The use of these chemicals is also far from ideal because they pose an intoxication risk to other animals including protected species, pets and humans.

The most important improvement in pest control technology was the development of anticoagulant rodenticides in the 1940s, with a second generation developed in the 1970s. These agents decrease blood clotting and their delayed effect means that rats consume a lethal dose before they show any symptoms of poisoning. With time, however, large populations of rats have acquired genetic resistance to these kinds of rodenticides (*Meerburg et al., 2014*), and third-generation anticoagulant rodenticides are currently under study (e.g., *Damin-Pernik et al., 2017*). Recently, integrated pest management strategies (focusing on long-term prevention or suppression of pest problems with minimum impact on human health and the wider environment) have been implemented to tackle rat infestations (*Flint et al., 2003*).

### Early history of research with the laboratory rat

The Norway rat is often considered the first mammal to have been domesticated for research purposes (*Richter, 1959*). Although some scientists point to the sporadic use of rats in experiments prior to 1850, the first known documented experiment conducted on these animals was a study of the effects of adrenalectomy published in 1856 in France (*Philipeaux, 1856*). In 1863, a study on the nutritional quality of proteins was conducted on mixed colored rats (*Savory, 1863*). The rat was first used in psychological studies by Adolph Mayer, a well-known American psychiatrist (*Logan, 1999*). After 1893, American neurologist Henry Herbert Donaldson started to use rats in biomedical experiments conducted at Chicago University (*Lockard, 1968*). When he took a post of the director of the Wistar Institute, he brought with him four pairs of albino rats that he then used in multidisciplinary studies conducted together with a large group of scientists. Donaldson intended to standardize the albino rat to create a universal model suited for biomedical research (*Lindsey and Baker, 2005*). Researchers at the Wistar Institute developed special breeding and reproduction techniques for rats. They designed special cages and entire buildings adapted specially for rat breeding. In 1912, the Wistar Institute began supplying laboratory rats to other research institutions (*Lindsey and Baker, 2005*).

The breeding colony established by Donaldson inspired his PhD student John Broadus Watson to conduct further experiments which resulted in ground-breaking discoveries in behavioral studies. In 1914, Watson published the book *Behavior: An Introduction to Comparative Psychology*, which became a major text in the field of animal psychology. His work was developed by Curt Paul Richter, who published numerous studies on topics such as domestication, stress, the biological clock and

**Table 1.** The most common stocks and strains of the laboratory rat.

| Name | Inbred/ outbred* | Coat color | Origine | Use and characteristics |
|---|---|---|---|---|
| Wistar | outbred | albino | The Wistar Institute, Philadelphia, Pennsylvania, USA (1906) | The most-popular general multi-purpose models. Studies of infectious diseases, aging and as a surgical model. |
| Wistar Han | outbred | albino | Zentralinstitute für Versuchstierzucht, Hannover, Germany | A general multi-purpose model, popular in preclinical safety assessments, and as an aging, oncological and surgical model. |
| Wistar Kyoto | outbred | albino | the Kyoto School of Medicine, Japan | Normotensive controls for the spontaneous hypertensive line, a depression and autism model. |
| Sprague Dawley | outbred | albino | The Sprague-Dawley farms, Madison, Wisconsin, USA (1925). Derived from a hybrid Hooded male and a female Wistar. | Behavioral studies and as models in obesity, oncology and surgical research. |
| Long Evans | outbred | hooded | The University of California, USA. Created by Herbert McClean Evans and Joseph Abraham Long (1915–1922). A result of crossbreeding albino females and wild males caught near the University. | Behavioral studies. Known for their docility and ease of breeding but prone to spontaneous seizures. |
| Brown Norway | inbred | pigmented | Derived from a pen-bred colony of wild-caught rats maintained by King and Aptekman at the Wistar Institute in the 1930s. The strain was created by Silvers and Billingham in 1958 (*Hedrich, 2000*). | Immunological and transplantation studies. Selected as the sequencing target in *Gibbs et al. (2004)*. |
| Lewis | inbred | albino | Developed by Margaret Lewis from the Wistar rats in the early 1950s | Enhanced susceptibility to many experimental inflammatory conditions, such as PGPS-induced arthritis, adjuvant-induced arthritis, collagen-induced arthritis, autoimmune encephalitis, autoimmune thyroiditis and enterocolitis (*Zhang, 2010*). Characterized by their docile behavior but relatively low fertility. |
| Zucker fatty rats | outbred | hooded | Developed by crossing the Sherman strain with the Meck stock 13M strain (*Kava et al., 1990*) | Most often used as a model of genetic obesity. Relatively insensitive to leptin due to a mutation in the long form of the leptin receptor (*van der Spek et al., 2012*). Characterized by hyperlipidemia, hypercholesterolemia and hyperinsulinemia (*Kava et al., 1990*). |
| Nude rats | inbred | albino hooded grey | The nude mutation first encountered in 1953 in an outbred colony of hooded rats at the Rowett Research Institute in Aberdeen, Scotland. The mutation reappeared independently in Aberdeen in 1977 and in New Zealand in 1979 (*Hanes, 2006*). Since than numerous new strains have been developed. For instance, a spontaneous mutation model isolated from a Crl:CD(SD) colony in Charles River in the late 1980s. | Characterized by almost complete absence of fur. Experimental models for a variety of immunological, surgical, infectious, transplant-related and oncological procedures. Uniquely capable of maintaining increased tumours without visible distress and enlarged body weight (*Hanes, 2006*). Also useful in wound healing and dermatology. |

*Inbred rat strains are created by brother-sister or parent-offspring mating for at least 20 generations. It produces almost genetically identical individuals (after 20 generations rats are homozygous at 98.7% of all alleles and the residual heterozygosity decreases as inbreeding continue; *Lohmiller and Swing, 2006*). Outbred rat stocks are developed from large colonies with males and females selected randomly from different breeding groups; stock animals are genetically different, which can represent inter-individual differences occurring in natural environment (*Lohmiller and Swing, 2006*; *Olson and Graham, 2014*).

adrenalectomy between 1919 and 1977 (*Lindsey and Baker, 2005*).

## Comparison with other animal models

Rats are often used in similar studies to mice (*Phifer-Rixey and Nachman, 2015*), though their larger size means they are more useful in some experiments, such as those involving surgery and imaging (*Jonckers et al., 2011*). Rat models are also considered more reliable than mouse models in the study of certain addictions

(*Vengeliene et al., 2014*), cancer immunotherapy (*Bergman et al., 2000*), and diabetes and related conditions (*Obrosova et al., 2006*). Some research areas in which rats are commonly used models now make more and more use of other animal models instead, such as the zebrafish (*Danio rerio*; *Parichy, 2015*; *Stewart et al., 2012*: *Kari et al., 2007*).

## Variety of strains and stocks

Numerous strains of laboratory rat have been created to ensure control over the genetic

## Box 3. Unanswered questions about the natural history of the laboratory rat.

Even though the rat is one of the oldest model organisms used in scientific studies, there are still many gaps in our knowledge about this species. By the same token, the common use of rats in scientific research generates new questions and doubts.

- Do the differences in morphology, physiology or behavior among rats of the same strain obtained from different breeders have a significant effect on the replicability of studies? What is the genetic variability within and between the laboratory populations of *R. norvegicus*? In other words, how stable and robust is the rat model based exclusively on the characteristics of a single strain?
- Nocturnal activity, a tendency to stay close to ground level, and a dominant sense of smell are all traits that rats likely share with the common ancestor of all mammals (*Finlay and Sengelaub, 1981*), but to what extent are the results obtained in studies conducted on rats also true of mammals in general and to what extent are they typical of rats only?
- The value of animal models in studying the effectiveness of, for instance, treatment strategies in clinical tests has remained controversial. To what extent can a single-species animal model, like the rat, accurately represent a process occurring in humans?
- Controversial aspects of using animals in scientific research, such as inflicting pain on animals, also raise questions. How often is it possible to use alternative methods and models for those experiments that have routinely used rats in the past?
- What is the genetic and epigenetic basis of their physiological and behavioral plasticity which allows rats to adapt to diverse environments? How will wild rat populations cope with rapid environmental changes, like climate change or the ubiquity of pharmacological substances in food and water?

variation in experimental subjects. However, the roots of the phylogenetic tree of the laboratory rat strains have not yet been established. Some researchers suggest several independent domestication pathways (e.g., *Festing, 1979*), but there is no consistent evidence to support this notion. More recent genetic studies based on the measurements of mutation rates in different parts of the rat genome have clarified the relationships between the different strains and led to a shared phylogenetic tree for most inbred strains (*Thomas et al., 2003*).

Based on their breeding history, laboratory rats may be broadly divided into outbred stocks and inbred strains (*Table 1*). The outbred stocks are usually used for general study purposes where homozygosity is not crucial and are well suited for behavioral studies. The inbred strains are used for researching issues related to genetic and phenotypic characteristics (*Sharp and Villano, 2012*). Rat models are also created in laboratories by means of electrical, pharmacological and surgical techniques that induce changes in the animals (e.g., *Calcutt, 2004*; *Teixeira and Webb, 2007*; *Relton and Weinreb, 2008*; *Obenaus and Kendall, 2009*).

Rat strains differ significantly in their morphology: their body weight and the size of internal organs may vary greatly, while the body length remains the same (e.g., *Reed et al., 2011*). For example, albino strains consistently exhibit impaired vision, while other strains appear to have the wild-type or even enhanced visual acuity (Prusky et a., 2002). Metabolism and behavior differ between certain strains as do the way these characteristics change with age (*Clemens et al., 2014*). Differences may also occur where social behaviors are concerned: for example, when play-fighting, juvenile Wistar rats initiate significantly fewer playful attacks than Fisher 344 rats (*Schneider et al., 2014*).

As many breeding colonies have been isolated for several decades, the inbred animals have different phenotypes than their counterparts bred elsewhere (e.g., *Goepfrich et al., 2013*). Environmental conditions and specific breeding settings lead to epigenetic differences, while several decades of breeding may result in a cumulation of mutations, which subsequently hinders the generalization of results even to the animals of the same strain (*Box 3*).

## Changes occurring in the process of laboratorization of *Rattus norvegicus*

### Morphological and physiological changes

The differences between laboratory rats and wild Norway rats were first noticed and described in the 1920s (*King and Donaldson, 1929*), when it was seen that laboratory rats differed from their wild counterparts in morphology and behavior after only 10 generations of inbreeding. In the second half of the 20th century, a series of morphological differences were spotted between the Wistar rats and trapped wild rats (*Richter, 1952*). The laboratory rats were smaller at maturity but did not differ significantly in their skeletal structure and teeth anatomy. The liver, heart, brain and adrenal glands were smaller, while the gonads and secondary sex organs developed at an earlier age (*Richter, 1952*). Domesticated female rats reached sexual maturity earlier and had bigger litters, which may indicate that domestication accelerated sexual development and increased reproductive success (*Clark and Price, 1981*). Domestication significantly affected their brain morphology too, the neocortex being the most markedly altered brain structure (*Welniak-Kaminska et al., 2019*). There are also significant differences in the circadian rhythm and out-of-nest activity between the laboratory and wild rats (*Stryjek et al., 2013*).

### Behavioral changes

Compared to their wild counterparts, laboratory rats show less interspecific aggression (*Barnett et al., 1979*). Defensive behaviors are also reduced, resulting in smaller reactions to both humans and conspecifics (*Blanchard et al., 1986*). Longitudinal studies of social behavior, such as play-fighting in juvenile rats, show that laboratory rats initiate more playful attacks and are more likely to defend themselves. Wild Norway rats are however more likely to use evasive actions to defend their nape than to wrestle with their partner (*Himmler et al., 2014*; *Himmler et al., 2013*).

In laboratory, where it is impossible to delineate separate territories, individual rats instead establish social hierarchies (*Adams and Boice, 1989*; *Blanchard et al., 1988*). Laboratory rats present a lower neophobia level (*Calhoun, 1963*; *Cowan, 1977*; *Tanaś and Pisula, 2011*), however early claims that laboratory rats exhibit lower food neophobia (*Barnett, 1958*;

*Mitchell, 1976*) were not replicated in a more recent study (*Modlinska et al., 2015*).

Both laboratory and wild rats explore their environments, but the response to a novel object in a familiar environment is less pronounced in wild subjects (*Tanaś and Pisula, 2011*). Domesticate rats seem to learn more quickly than wild rats (*Price, 1972*), tending to perform better in laboratory learning paradigms (*Boice, 1981*).

Wild rats have a broad repertoire of swimming-related behaviors, while laboratory rats are reluctant to swim (*Stryjek et al., 2012*). Wild rats build more complex and more durable tunnels and, unlike their laboratory cousins, inhabitable underground burrows (*Stryjek et al., 2012*).

## Impact of domestication on research and research results

Differences between laboratory rats and wild rats had previously prompted several scientists to question the legitimacy of generalizing the results of studies conducted on laboratory rats to the species as a whole, or other organisms (*Beach, 1950*; *Lockard, 1968*). Yet comparative studies have shown that domestication rarely modifies an animal's behavioral repertoire to any significant extent (*Price, 1999*; *Stryjek et al., 2012*; *Modlinska et al., 2015*). Instead, most changes tend to affect the frequencies of certain behaviors, or the thresholds at which a stimulus will trigger a response.

Some features of domestication have also unintentionally increased the utility of rats as a model organism. For instance, the laboratory rats' reluctance to swim and their determined attempts to get out of water are crucial to the Water Morris Test, a popular protocol in the study of memory and learning (cf. *Whishaw and Pasztor, 2000*).

## Attempts to recreate new laboratory rat populations from wild colonies

Several researchers aware of the problems arising from the domestication of the rat conducted experiments on wild Norway rats and comparative studies of both lines. Samuel Anthony Barnett, the author of the classic text "*The Rat: A Study in Behaviour*" (first published in 1963), caught wild rats and studied them in his laboratory for decades since 1950s, and in the process developed several techniques for handling them (*Barnett, 2009*). Beginning in 1970s, Bennett G

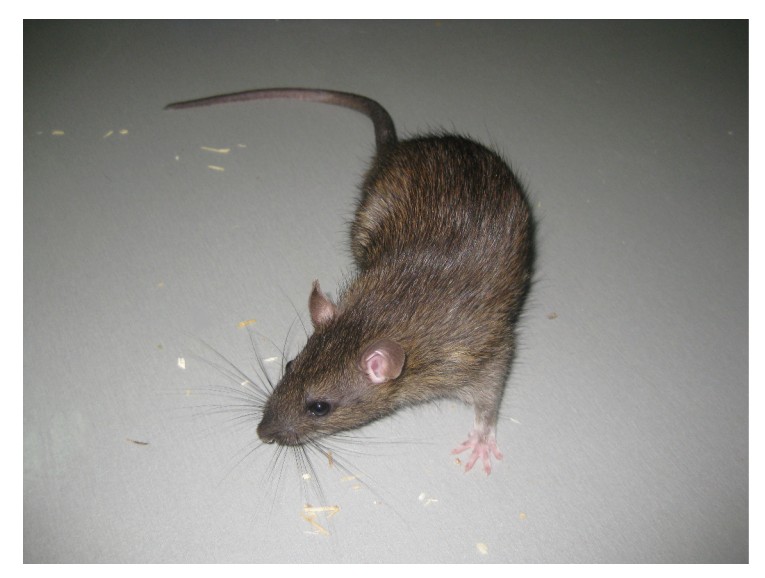

**Figure 1.** A laboratory-bred wild rat. *R. norvegicus* is a relatively small rodent with a brown fur and sparsely haired tail. Its head is stout with a pointed muzzle and darkly pigmented, slightly bulging eyes. Characteristic of all rodents, rats have large and continuously growing front teeth. The durable enamel on the front surface of these teeth contains an iron-based pigment, which gives them an orange color. This individual belongs to the Warsaw Wild Captive Pisula Stryjeck (WWCPS) colony in Poland. Image credit: Klaudia Modlinska and Rafał Stryjek.

Galef also extensively studied wild Norway rats with a specific focus on their feeding behaviors (e.g., *Galef and Clark, 1971*), and Robert J Blanchard spent many years investigating the defensive and aggressive behaviors of these animals (e.g., *Blanchard et al., 1986*).

Jaap Koolhaas also conducted experiments with wild caught Norway rats in the late 1990s (*Koolhaas et al., 1999*). He studied stress and aggression, and the wild rats were particularly well suited for those experiments due to their poor adaptation to the laboratory setting and their emotional constitution. His work on wild rats resulted in the creation of a wild line of *R. norvegicus* – the Wild-type Groningen rats.

In 2006, a new laboratory colony of wild Norway rats was set up in Poland (*Stryjek and Pisula, 2008*). The new line was named WWCPS, which short for Warsaw Wild Captive Pisula Stryjek (*Figure 1*). In order to prevent the development of domestication features in the breeding colony and maintain the animals' 'wild' genetic status, the colony was systematically enlarged with captured rats in various locations. Since it was established, comparative studies involving rats from this colony have added to the list of known differences between wild rats and laboratory rat lines (*Stryjek et al., 2012*; *Modlinska et al., 2015*; *Himmler et al., 2014*; *Himmler et al., 2013*; etc.).

It is important, however, to note that wild rats are not easily handled or manipulated. The fact that these animals are less suited to a laboratory setting can impact the results obtained from them. Wild rats in a laboratory have a higher level of stress hormones in their blood plasma than domesticated laboratory rats; they also exhibit stronger responses to emotional stressors and novel objects (*Naumenko et al., 1989*; *Plyusnina et al., 2011*; *Koizumi et al., 2019*). These factors must be taken into consideration when interpreting results and may constrain the kind of studies that are feasible using wild rats. Before conducting experiments with wild individuals, researchers may need to develop special procedures that better approximate the natural conditions of these animals (i. e., that have "high ecological validity"). Efforts must be made to reduce the stress involved in the breeding and experimental manipulations, as it may affect rat welfare. Nevertheless, studies on wild animals, that have not been subjected to the domestication process, could help the community to assess the generality or specificity of results obtained with laboratory lines. The fact that wild rats show more variability between individuals with regard to many biological traits may also be useful when studying the impact of various stimuli (e.g., environmental changes) on such complex and variable populations. Such experiments would be difficult to achieve using standardized laboratory strains.

## Conclusion

Many of the traits that make Norway rats a pest in the wild are the same traits that have contributed to its success as a model organism. Nevertheless, the domestication of the rat for research purposes has also resulted in significant changes. Rather than viewing the rat as a simple model, a "pest" or a "pet", it is important to recognize it as a complex mammal in its own right, and one that is highly adapted to its environment (*Burn, 2008*). Research on rats in the laboratory will be benefited by researchers who understand the animals they are working with; this includes having an appreciation of the rat's natural history.

**Klaudia Modlinska** is at the Institute of Psychology, Polish Academy of Sciences, Warsaw, Poland

kmodlinska@psych.pan.pl

https://orcid.org/0000-0002-2161-9019

**Wojciech Pisula** is at the Institute of Psychology, Polish Academy of Sciences, Warsaw, Poland

*Author contributions:* Klaudia Modlinska, Writing - original draft, Writing - review and editing, Conceptualization, Investigation, Methodology; Wojciech Pisula, Writing - original draft, Conceptualization, Investigation, Methodology

*Competing interests:* The authors declare that no competing interests exist.

## Funding

| Funder | Grant reference number | Author |
| --- | --- | --- |
| Narodowe Centrum Nauki | UMO-2015/19/D/HS6/00781 | Klaudia Modlinska |

The funders had no role in study design, data collection and interpretation, or the decision to submit the work for publication.

## Decision letter and Author response

Decision letter https://doi.org/10.7554/eLife.50651.sa1
Author response https://doi.org/10.7554/eLife.50651.sa2

## Additional files

### Data availability

No data was generated as part of this work.

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
