## [Decision Letter]

**Acceptance summary:**

The "lab rat" is a classical model organism but less is known about its former life in the wild. This wide-ranging review gives an interesting introduction to laboratory rats from an ecological perspective, discussing how they compare with their wild counterparts, and covering the history of the domestication of this model species. The revisions have strengthened the article and it will soon make a welcome addition to our collection on the natural history of model organisms. This work should be of special interest to researchers who study rats in the laboratory.

**Decision letter after peer review:**

Thank you for submitting your article "The Natural History of Model Organisms: The Norway rat, from an obnoxious pest to a laboratory pet" for consideration by *eLife*. Your article has been reviewed by two peer reviewers and the evaluation has been overseen by two Features Editors at *eLife* (Stuart King and Peter Rodgers). The following individual involved in review of your submission has agreed to reveal their identity: Amelie Desvars.

The reviewers have discussed the reviews with one another and the Associate Features Editor has drafted this decision to help you prepare a revised submission.

Summary:

This essay is being considered as part of a series of articles on "The Natural History of Model Organisms":https://elifesciences.org/collections/8de90445/the-natural-history-of-model-organisms. Each article should explain how our knowledge of the natural history of a model organism has informed recent advances in biology, and how understanding its natural history can influence/advance future studies.

The "lab rat" is perhaps the archetypical model organism and would thus make a welcome and interesting addition to this collection of articles. The paper is also timely. Rats remain a key laboratory animal for much research, especially in the behavioral and neural sciences, yet there is a nagging suspicion about the consequences of over a hundred years of domestication.

This wide-ranging review explores the history of laboratory rats, their uses and how they compare with their wild counterparts. The conclusion is that laboratory rats retain sufficient physiological and behavioral characteristics of wild rats to be suitable as animal models for many questions. It also concludes that all strains, or stocks thereof, including wild rats, have to be thoughtfully matched to the research question being asked. This will be a valuable resource in guiding researchers to use rats as animal models more effectively, nevertheless revisions are needed to strengthen the article.

Essential revisions:

1) Structure of the article

Overall, the article is comprehensive and well-researched, with many examples. It would, however, benefit from editing to make the text more succinct. Restructuring would also help its ideas to flow more fluidly and make its central message/conclusion clearer.

Below are some general suggestions as to how this could be achieved.

- Introduction

Taking inspiration from the title, the Introduction could be restructured into three, short paragraphs (max 150 words each). The first paragraph could briefly introduce wild rats as one of the most important vertebrate pest species (with risks to public health, animal health, wildlife, agriculture and infrastructures), and explain how they are widely disliked by the public. The second paragraph could then contrast this by describing laboratory rats as a popular model organism with a long history in research. The third paragraph should highlight the concerns about the "laboratorisation" of rats and briefly describe the objectives or central theme for the rest of the article. The third paragraph is the most critical one in the Introduction. All three paragraphs should offer a high level perspective, with more detail given later in the main text.

- Main text

Most sections would benefit from being more concise. For many sections, the word count could be cut by about a third without reducing the scope. The reviewers felt that some topics were discussed in inappropriate sections (i.e. "diet" is currently combined with "behaviour", and "reproduction" is included under "physical traits".

The section on "Natural history" should focus on the origin, evolution, phylogenetics, biogeography of wild rats. The section on Ecology could be a sub-section of this section, and should discuss the distribution of rats in cities more.

Difference between lab and wild rats are currently described in three consecutive sections: "Laboratorisation of *R. norvegicus*", "New laboratory rat populations recreated from wild colonies" and "Comparative studies on wild and laboratory rats". These three sections could be revised and restructured to describe i) the changes that occurred when wild rats were domesticated for use in the lab, ii) how this subsequently limited the usefulness of lab rats for some research, and iii) how researchers try to overcome these limitations by creating new lab rat populations from wild colonies (with mentions of the advantages and limitations of these new stocks).

- Conclusion

This also needs to be condensed and should avoid introducing too many new concepts that were not discussed in the article.

- Box 1: Disease and pest control

This also needs to offer a high level perspective and can be cut back to avoid too much detail on specific examples of diseases spread by rats. It would be good to instead briefly cover public opinion of rats (as dirty animals, living in sewage and feeding on garbage), and the impact of rats on crops, infrastructures and endangered wildlife. It would be interesting to mention here how humans have tried for centuries to eliminate, or at least control, rat populations, and that research on rodenticide resistance involves both lab and wild rats.

2) Figures and tables

The current manuscript has a photo as one figure and three boxes to discuss specific topics that would otherwise disrupt the flow of the text. The reviewers felt that the authors should consider moving some of the details currently written in the text into new tables or figures. For example, the information about the systematic groups in the genus *Rattus* (subsection “Natural history”, second paragraph) should be removed from the text, and presented as a table, or perhaps as a figure with a phylogenetic tree. A map could help the author explain how the Norway rat colonized different geographic regions (see the last two paragraphs of the aforementioned subsection), and the information displayed in Box 2, "The most common stocks and strains of the laboratory rat", would be more easily read if it was presented in a table.

3) Species common name

It would also be interesting if the authors could briefly explain why this rat is called the "Norway rat" when its origins are thought to be in Asia. A few sentences in the appropriate section would likely satisfy a reader's curiosity.

On a related point, it would be good if the article could also list the other common names, besides brown rat, for completeness – i.e. sewer rat, water rat, city rat, common rat – but then continue to use one name, i.e. Norway rat, throughout the rest of the article to avoid confusing unfamiliar readers.

4) Wild or lab rats?

In some sections, especially under the heading Characteristics of the species", it is unclear whether the text refers to wild rats, laboratory strains or both. Please go through the text and make this clearer. It may help if that specific section is renamed "Characteristics of wild Norway rats, and any comparison to laboratory rats is made explicit, or saved for a later section.

5) References

Several statements need support references from the literature while some references should be updated.

6) Part of a collection

Lastly, since this article is part of a series that has already covered 12 other model organisms (including two other rodents), it would be good if the authors could do more to highlight similarities/differences between rats and any of the other model organisms in the series. For example, when discussing life history traits that make rats a good choice for a model organism, it'd be interesting to note other models that have similar traits, and cite the relevant articles already in the collection to help readers see the connections [https://elifesciences.org/collections/8de90445/the-natural-history-of-model-organisms].

---

## [Author Response]

Essential revisions:1) Structure of the articleOverall, the article is comprehensive and well-researched, with many examples. It would, however, benefit from editing to make the text more succinct. Restructuring would also help its ideas to flow more fluidly and make its central message/conclusion clearer.

Thank you for this comment. The manuscript has been restructured and edited to make it more concise.

Below are some general suggestions as to how this could be achieved. The Associate Features Editor will contact you separately with more specific edits.- IntroductionTaking inspiration from the title, the Introduction could be restructured into three, short paragraphs (max 150 words each). The first paragraph could briefly introduce wild rats as one of the most important vertebrate pest species (with risks to public health, animal health, wildlife, agriculture and infrastructures), and explain how they are widely disliked by the public. The second paragraph could then contrast this by describing laboratory rats as a popular model organism with a long history in research. The third paragraph should highlight the concerns about the "laboratorisation" of rats and briefly describe the objectives or central theme for the rest of the article. The third paragraph is the most critical one in the Introduction. All three paragraphs should offer a high level perspective, with more detail given later in the main text.

Following your advice, we have rewritten the Introduction section. As you suggested, we have divided the section into three paragraphs, and in the first part we have presented the rat as a nuisance, in the second part we have considered the rat as a laboratory model, and in the last part we have briefly described the controversy around using the domesticated form of the species.

- Main textMost sections would benefit from being more concise. For many sections, the word count could be cut by about a third without reducing the scope. The reviewers felt that some topics were discussed in inappropriate sections (i.e. "diet" is currently combined with "behaviour", and "reproduction" is included under "physical traits". As mentioned above, the Associate Features Editor will contact you separately with specific edits to help address these issues.

The manuscript has been restructured and edited to make it more succinct. The word count has been cut substantially. The different sections have been reordered.

The section on "Natural history" should focus on the origin, evolution, phylogenetics, biogeography of wild rats. The section on Ecology could be a sub-section of this section, and should discuss the distribution of rats in cities more.

The Ecology section has been replaced and now follows the Natural History section. Both sections have been revised.

Difference between lab and wild rats are currently described in three consecutive sections: "Laboratorisation of R. norvegicus", "New laboratory rat populations recreated from wild colonies" and "Comparative studies on wild and laboratory rats". These three sections could be revised and restructured to describe i) the changes that occurred when wild rats were domesticated for use in the lab, ii) how this subsequently limited the usefulness of lab rats for some research, and iii) how researchers try to overcome these limitations by creating new lab rat populations from wild colonies (with mentions of the advantages and limitations of these new stocks).

The sections you mentioned above have been rewritten and restructured accordingly.

- ConclusionThis also needs to be condensed and should avoid introducing too many new concepts that were not discussed in the article.

The conclusion section has been shortened, and it is now only a brief summary.

- Box 1: Disease and pest controlThis also needs to offer a high level perspective and can be cut back to avoid too much detail on specific examples of diseases spread by rats. It would be good to instead briefly cover public opinion of rats (as dirty animals, living in sewage and feeding on garbage), and the impact of rats on crops, infrastructures and endangered wildlife. It would be interesting to mention here how humans have tried for centuries to eliminate, or at least control, rat populations, and that research on rodenticide resistance involves both lab and wild rats.

The section has been revised and the first paragraph shorted as per reviewer’s comments. The references have been updated and missing information added.

2) Figures and tablesThe current manuscript has a photo as one figure and three boxes to discuss specific topics that would otherwise disrupt the flow of the text. The reviewers felt that the authors should consider moving some of the details currently written in the text into new tables or figures. For example, the information about the systematic groups in the genus Rattus (subsection “Natural history”, second paragraph) should be removed from the text, and presented as a table, or perhaps as a figure with a phylogenetic tree. A map could help the author explain how the Norway rat colonised different geographic regions (see the last two paragraphs of the aforementioned subsection), and the information displayed in Box 2, "The most common stocks and strains of the laboratory rat", would be more easily read if it was presented in a table.

As you suggested, we have transferred the information about the systematic groups in the genus *Rattus* to a separate box. We have also moved the description of methods for creating rat models in the laboratory to Box 2 (the most common stocks and strains of the laboratory rat). The presentation of stocks and strains in Box 1 has also been rewritten and reformatted into a table.

3) Species common nameIt would also be interesting if the authors could briefly explain why this rat is called the "Norway rat" when its origins are thought to be in Asia. A few sentences in the appropriate section would likely satisfy a reader's curiosity.On a related point, it would be good if the article could also list the other common names, besides brown rat, for completeness – i.e. sewer rat, water rat, city rat, common rat – but then continue to use one name, i.e. Norway rat, throughout the rest of the article to avoid confusing unfamiliar readers.

A brief explanation of the origin of the name "Norway rat" has been added to the Natural History section. The commonly used names have been listed in the Introduction.

4) Wild or lab rats?In some sections, especially under the heading "Characteristics of the species", it is unclear whether the text refers to wild rats, laboratory strains or both. Please go through the text and make this clearer. It may help if that specific section is renamed "Characteristics of wild Norway rats", and any comparison to laboratory rats is made explicit, or saved for a later section.

We have revised the characteristics of the species to make sure it only described the characteristics of the wild rat. Changes that had occurred during the domestication process have been presented in a separate section "Changes occurring in the process of laboratorisation of *Rattus norvegicus".*

5) ReferencesSeveral statements need support references from the literature while some references should be updated.

The references have been revised and updated according to the reviewer’s suggestions.

6) Part of a collectionLastly, since this article is part of a series that has already covered 12 other model organisms (including two other rodents), it would be good if the authors could do more to highlight similarities/differences between rats and any of the other model organisms in the series. For example, when discussing life history traits that make rats a good choice for a model organism, it'd be interesting to note other models that have similar traits, and cite the relevant articles already in the collection to help readers see the connections [https://elifesciences.org/collections/8de90445/the-natural-history-of-model-organisms].

A brief comparison with other animal models has been presented in a separate section "Comparison with other animal models", and relevant articles from the series have been cited.